# Phytochemicals of *Withania somnifera* as a Future Promising Drug against SARS-CoV-2: Pharmacological Role, Molecular Mechanism, Molecular Docking Evaluation, and Efficient Delivery

**DOI:** 10.3390/microorganisms11041000

**Published:** 2023-04-12

**Authors:** Suaidah Ramli, Yuan Seng Wu, Kalaivani Batumalaie, Rhanye Mac Guad, Ker Woon Choy, Ashok Kumar, Subash C. B. Gopinath, Md. Moklesur Rahman Sarker, Vetriselvan Subramaniyan, Mahendran Sekar, Neeraj Kumar Fuloria, Shivkanya Fuloria, Suresh V. Chinni, Gobinath Ramachawolran

**Affiliations:** 1Department of Pharmacy, Hospital Sultanah Nur Zahirah, Kuala Terengganu 20400, Malaysia; suaidahramli97@gmail.com; 2Centre for Virus and Vaccine Research, School of Medical and Life Sciences, Sunway University, Subang Jaya 47500, Malaysia; 3Department of Biological Sciences, School of Medical and Life Sciences, Sunway University, Subang Jaya 47500, Malaysia; 4Department of Biomedical Sciences, Faculty of Health Sciences, Asia Metropolitan University, Johor Bahru 81750, Malaysia; kalaivanibatumalaie@amu.edu.my; 5Department of Biomedical Science and Therapeutics, Faculty of Medicine and Health Science, Universiti Malaysia Sabah, Kota Kinabalu 88400, Malaysia; rhanye@ums.edu.my; 6Department of Anatomy, Faculty of Medicine, Universiti Teknologi MARA, Sungai Buloh 47000, Malaysia; choykerwoon@uitm.edu.my; 7Department of Internal Medicine, Division of Pulmonary, University of Kansas Medical Center, 3901 Rainbow Boulevard, Kansas City, KS 66160, USA; akb.bits@gmail.com; 8Centre of Excellence (CoE), Faculty of Chemical Engineering & Technology & Micro System Technology, Universiti Malaysia Perlis (UniMAP), Arau 02600, Malaysia; subash@unimap.edu.my; 9Institute of Nano Electronic Engineering, Universiti Malaysia Perlis (UniMAP), Kangar 01000, Malaysia; 10Department of Pharmacy, State University of Bangladesh, 77 Satmasjid Road, Dhanmondi, Dhaka 1205, Bangladesh; 11Health Med Science Research Network, 3/1, Block F, Lalmatia, Dhaka 1207, Bangladesh; 12Pharmacology Unit, Jeffrey Cheah School of Medicine and Health Sciences, MONASH University, Jalan Lagoon Selatan, Bandar Sunway, Subang Jaya 47500, Malaysia; subramaniyan.vetriselvan@monash.edu; 13Department of Pharmacology, School of Medicine, Faculty of Medicine, Bioscience and Nursing, MAHSA University, Subang Jaya 42610, Malaysia; 14School of Pharmacy, Monash University Malaysia, Bandar Sunway, Subang Jaya 47500, Malaysia; mahendran.sekar@monash.edu; 15Centre of Excellence for Biomaterials Engineering & Faculty of Pharmacy, AIMST University, Bedong 08100, Malaysia; neerajkumar@aimst.edu.my; 16Center for Transdisciplinary Research, Department of Pharmacology, Saveetha Institute of Medical and Technical Sciences, Saveetha Dental College and Hospitals, Saveetha University, Chennai 600077, India; 17Faculty of Pharmacy, AIMST University, Semeling, Bedong 08100, Malaysia; shivkanya_fuloria@aimst.edu.my; 18Department of Biochemistry, Faculty of Medicine, Bioscience, and Nursing, MAHSA University, Jenjarom 42610, Malaysia; venkatasuresh@mahsa.edu.my; 19Department of Periodontics, Saveetha Institute of Medical and Technical Sciences, Saveetha Dental College and Hospitals, Chennai 600077, India; 20Department of Foundation, RCSI & UCD Malaysia Campus, No 4, Jalan Sepoy Lines, Georgetown 10450, Malaysia

**Keywords:** COVID-19, SARS-CoV-2, *Withania somnifera*, Ashwagandha, Ayurveda, Rasayana

## Abstract

Coronavirus disease (COVID-19) has killed millions of people since first reported in Wuhan, China, in December 2019. Intriguingly, *Withania somnifera* (WS) has shown promising antiviral effects against numerous viral infections, including SARS-CoV and SARS-CoV-2, which are contributed by its phytochemicals. This review focused on the updated testing of therapeutic efficacy and associated molecular mechanisms of WS extracts and their phytochemicals against SARS-CoV-2 infection in preclinical and clinical studies with the aim to develop a long-term solution against COVID-19. It also deciphered the current use of the in silico molecular docking approach in developing potential inhibitors from WS targeting SARS-CoV-2 and host cell receptors that may aid the development of targeted therapy against SARS-CoV-2 ranging from prior to viral entry until acute respiratory distress syndrome (ARDS). This review also discussed nanoformulations or nanocarriers in achieving effective WS delivery to enhance its bioavailability and therapeutic efficacy, consequently preventing the emergence of drug resistance, and eventually therapeutic failure.

## 1. Introduction

The world is currently struggling to save people from a newly emerged coronavirus disease (COVID-19) that started at the end of 2019, noticed in a non-human source in Wuhan province of China. The World Health Organization (WHO) has announced that COVID-19 is a worldwide pandemic, and several countries have encountered a great loss in human health and economy [1]. The federal and non-federal governments are urging scientists to produce an efficient strategy against severe acute respiratory syndrome coronavirus 2 (SARS-CoV-2) virus. There are different opinions and concepts on the origin of this coronavirus as it is not new for us because we have faced similar issues in the past. Although the world has experienced similar pandemic problems, such as the Middle East respiratory syndrome (MERS), severe acute respiratory syndrome (SARS), the plague, Swine-flu, Spanish flu, Ebola, Zika, and Nipah, COVID-19 caused by the SARS-CoV-2 behaves uniquely and differently. The present situation of COVID-19 urges scientists to generate a rapid remedy for normalizing the infected patients to routine life. Due to the lack of immediate medical solutions, the world widely agreed that the strategies for preventing disease spread are to practice social distancing and perform simultaneous screening through confirmation tests a step ahead of generating anti-SARS-CoV-2 drugs.

Currently, 517.65 million cases and 6.3 million deaths have been reported worldwide as of 15 May 2022 [2]. Although the rising cases of COVID-19 have been controlled with vaccination and confirmed screening tests (i.e., reverse transcription polymerase chain reaction (RT-PCT) and antigen tests), there is a growing number of severe cases of COVID-19 with higher mortalities among elderly patients [3]. Moreover, combating SARS-CoV-2 has been a tremendous global challenge, especially the re-emergence of SARS-CoV-2 in the near future with the seasonal variation that requires an effective strategy. Similar to other pandemics, it has been shown that the production of effective drugs could provide a long-term solution for COVID-19 [4]. In this regard, herbal medicine could be explored in developing effective drugs, as it may provide benefits in terms of availability, economy, and ease of access.

Natural products offer advantages in their structural complexity and vast scaffold diversity, which might be beneficial in tackling protein-protein interactions [5]. Moreover, many herbal products were proven to exert antiviral effects, including coronavirus, dengue virus, hepatitis virus, and human immunodeficiency virus (HIV) [6]. Thus far, several studies have investigated the potential of herbal medicine to prevent and treat COVID-19 by inhibiting SARS-CoV-2 pathogenesis [7,8].

*Withania somnifera* (WS), also known as Ashwagandha, has been widely used in Indian traditional medicine, Chinese traditional medicine, and the Unani system of medicine for thousands of years to promote vitality and youthfulness, besides improving physical and mental well-being [9]. The ancient use of traditional medicine should be explored as it might bear the imprint of the flourishing healthcare system inherited by mankind [10]. WS has been recognized for numerous pharmacological functions, including anticancer, antidiabetic, anti-inflammatory, neuroprotective, and cardioprotective, and its beneficial roles in dermatological diseases and osteoporosis [11,12,13]. Intriguingly, WS has shown promising antiviral properties against SARS-CoV-2, contributed by its various phytochemicals, including the frequently reported withaferin A, withanolide, withanolide A, withanolide B, withanolide R, 2,3-dihydrowithaferin A, withanone, withanoside IV, withanoside V, withanoside X, quercetin glucoside, somniferin, etc.

Therefore, the present review aims to provide recent updates and insights on the current issues of COVID vaccines and medications that focus on the pharmacological role of WS and its phytochemicals against SARS-CoV-2, as evidenced by preclinical and clinical studies. Additionally, it also summarizes the associated molecular mechanisms and key molecules involved. Furthermore, in silico molecular docking evaluation of WS toward SARS-CoV-2 target proteins is comprehensively focused on. Lastly, a potential delivery strategy to enhance active targeting of WS phytochemicals against SARS-CoV-2 is also discussed. All of which intend to present potential future promising drugs of WS phytochemicals for subsiding SARS-CoV-2 infection.

## 2. Methodology for Literature Search/Selection

The scientific articles and reviews were searched and screened from different databases, including PubMed, ScienceDirect, Google Scholar, National Center for Biotechnology Information (NCBI), and ResearchGate from January 2020 until May 2022 to identify the most related and recent articles by using the Medical Subject Heading (MeSH) and relevant keywords, such as “*Withania somnifera*”, “Ashwagandha”, “Ayurveda”, “Rasayana”, “SARS-CoV-2”, and “COVID-19”. The research articles published only in the English language were screened. In analyzing the possible interactions with SARS-CoV-2 and mechanisms of actions in combating COVID-19, in silico tools have been widely utilized with extensive virtual screening of the potential phytochemicals of WS involved [14].

The brief PRISMA flowchart (Figure 1) summarizes the process of selecting the relevant studies to prepare this review based on the previous reported guidelines [15,16].

## 3. Overview of SARS-CoV-2

SARS-CoV-2 has been identified as a betacoronavirus with 45–90% similarity as compared with that of SARS-CoV but is different in terms of the S gene that interacts with host cells to encode the S protein [17,18,19]. SARS-CoV-2 is a 30 kB single-stranded positive with 5′-cap and 3′-poly(A) tail RNA virus, allowing it to perform as a functional mRNA for replication-translation events [20]. The replicase genes referred to as Open Reading Frame 1a and ab (ORF1ab) may encode nonstructural proteins (NSPs) referring to pp1a and pp1ab polyproteins, occupying two-thirds of the viral genomic area, which is the first 20 kb lying downstream to the 5′-end [20]. NSP pp1a has 10 non-synonymous serine residues expressed as NSP1 to NSP11, while NSP pp1ab has 16 non-synonymous serine residues expressing NSP12 to NSP16 [20,21]. Meanwhile, four structural genes (i.e., S, E, M, and N) are encoded in the remaining 10 kb region preceding the 3′-end. ORF3a, ORF3d, ORF6, ORF7a, ORF7b, ORF8, ORF9b, ORF14, and ORF10 genes encode nine auxiliary proteins in addition to the structural genes [20]. The SARS-CoV-2 genome structure is further illustrated in Figure 2.

SARS-CoV-2 is prone to genetic evolution, resulting in the formation of the worldwide prevalent D614G variation, which is linked with greater transmissibility [22]. Numerous variants of SARS-CoV-2 have been described since then, some of which are considered variants of concern (VOCs) due to their potential to increase transmissibility or virulence, decrease neutralization by antibodies obtained through natural infection or vaccination; the ability to evade detection; or a decrease in therapeutic or vaccination effectiveness. In late December 2020, the United Kingdom (UK) revealed the discovery of a new SARS-CoV-2 VOC, the B.1.1.7 lineage, commonly known as the Alpha variant or GRY (formerly GR/501Y.V1) [23,24]. It is 43–82% more transmissible than pre-existing SARS-CoV-2 variations, thus emerging as the dominant SARS-CoV-2 variant in the UK [25]. At the end of December 2020, the B.1.1.7 variant that has 17 mutations was recorded in the United States, resulting in the S protein’s increased binding affinity to ACE2 receptors, facilitating viral attachment and entry into host cells [25,26,27].

A second SARS-CoV-2 variant (B.1.351) is commonly known as the Beta variant or GH501Y. V2, with numerous S gene mutations, was first found in October 2020 in South Africa and reported by the health department of the country in December 2020, resulting in the second wave of COVID-19 infection [25,28]. The B.1.351 variation has nine mutations in the S protein. Three of which are situated in the RBD, which further enhance the binding affinity of B.1.351 to ACE receptors [26,28,29]. The third VOC is the P.1 variant, also known as the Gamma variant or GR/501Y.V3. It was first discovered in Brazil in December 2020, followed by reports in the US in January 2021 [30]. This variant was repoetd to have 11 S protein mutations. It is further discovered that three mutations (i.e., L18F, K417N, and E484K) in the RBD are homologous to the B.1.351 variation [31,32]. The fourth VOC is B.1.617.2, known as the Delta variant. It was first discovered in December 2020 in India and was implicated in its devastating second wave of COVID-19 infection in April 2021. This SARS-CoV-2 variation was discovered for the first time in the USA in March 2021 [30]. The B.1.617 variant carries 10 mutations in the S protein [30].

## 4. Pathogenesis and Clinical Manifestations of SARS-CoV-2

### 4.1. SARS-CoV-2 Pathogenesis

Generally, SARS-CoV-2 is transmitted primarily through infected respiratory droplets that come into contact, either directly or indirectly, with nasal, conjunctival, or oral mucosa [33]. Additionally, the target host receptors are primarily present in the epithelium respiratory tract [33]. S protein attaches to the ACE2 receptor on the cell surface when a human is infected with SARS-CoV-2 before priming by a host proteinase called transmembrane serine protease 2 (TMPRSS2) to promote its entry into host cells, as demonstrated in Figure 3 [34,35]. After hijacking the cell’s replication and translation machinery in alveolar epithelial cells, SARS-CoV-2 consumes stored energy and chemicals in host cells, such as nucleotides, amino acids, and lipids, resulting in tremendous viral replication in a short time and ultimately leading to cellular collapse.

On the other hand, bodily cells defend themselves against viruses by developing strong immune systems. For instance, macrophages phagocytose whole viral particles and hydrolyze them in lysosomes with numerous hydrolytic enzymes [36]. If SARS-CoV-2 evades the immune elimination and releases its RNA, it will be identified and recognized by the receptor toll-like receptor (TLR) 7, which is found on the endosomal membrane [37]. Multiple proteins are recruited to form a complex after viral RNA activates TLR7, which consequently increases the transfer of transcription factors, such as nuclear factor-kappa B (NF-kB) and interferon regulatory factor 7 (IRF7) to the nucleus, triggering the production of pro-inflammatory cytokines [38]. These pro-inflammatory cytokines can cause the immune system of host cells to eliminate SARS-CoV-2 from infecting host cells. However, an overactive immune system causes cytokines to infiltrate the lung tissue and thus induce a pathology of cytokine storm, which can lead to acute respiratory distress syndrome (ARDS), multiple organ failure (MOF), and even death [39]. Other than alveolar epithelial cells, human monocytes, and macrophages also express ACE2 receptors and can be infected with SARS-CoV-2 directly, thus worsening the transcription of pro-inflammatory genes contributing to the development of COVID-19. Furthermore, according to recent studies, the S protein of the SARS-CoV-2 can bind to the neuropilin-1 (NRP1) receptor on host cells and subsequently boost the SARS-CoV-2 ability to infect host cells [40]. For better insight, the SARS-CoV-2’s pathogenesis on host lung epithelial cells is depicted in Figure 4.

### 4.2. Clinical Manifestations of SARS-CoV-2

Although most of the COVID-19 symptoms are minor in most patients in general, some are mild at first and worsen suddenly, leading to multiple MOF and subsequently death, which may be caused by cytokine storms, which usually occur in critical illness patients [41]. Additionally, patients with severe SAR-CoV-2 infection may develop ARDS, which typically occurs around one week after the beginning of symptoms. A SARS-CoV-2 infection has a median incubation time of 5.1 days, and most patients exhibit symptoms within 11.5 days after infection [42]. The clinical spectrum of COVID-19 ranges from asymptomatic or paucisymptomatic infection to severe respiratory failure necessitating mechanical ventilation, septic shock, and MOF. Among them, only 17.9–33.3% of infected patients remain asymptomatic [43,44]. The great majority of symptomatic patients are presented with fever, cough, and shortness of breath, with a sore throat, anosmia, dysgeusia, anorexia, nausea, malaise, myalgias, and diarrhea being less prevalent.

## 5. COVID-19 Vaccines against SARS-CoV-2

Vaccines appear to be an intriguing option for dealing with the pandemic with the purpose to create primary body defense against SARS-CoV-2. Nonetheless, currently limited human evidence is available for these vaccines. Although vaccines based on SARS-CoV-2 proteins stimulate both humoral and cell-mediated immune responses, they require adjuvant to activate immunological responses, and S protein is difficult to express, which is expected to affect production yields and the number of doses available [45]. To overcome these constraints, mRNA-based vaccines, such as lipid vesicles (liposomes), have been developed by Abogn (China), CureVac (Germany), and Moderna (US). This vaccine is based on delivering genetic information for the antigen rather than the antigen itself, and the antigen is then produced in the vaccinated individual’s cells [46].

Apart from this, RNA must be delivered in a variety of ways to enter a human cell. Since frozen storage is required, large-scale production and long-term storage stability are also unknowns. Because vaccines are given by injection, they are unlikely to produce substantial mucosal immunity [47]. Naked DNA-based vaccines, on the other hand, such as naked DNA plasmids plus electroporation Inovio (US) and Genexine (Korea) [46], frequently have low immunogenicity and must be provided via delivery devices to be effective. This issue requires the employment of delivery methods, such as electroporators, hence restricting their application [47]. Vaccines that use a viral vector, either engineered non-replicating virus vectors (replication-incompetent vectors), engineered replicating virus vectors (replication-competent vectors), or inactivated virus vectors, are also being studied in clinical trials around the world. Chimpanzee adenovirus (AstraZeneca) and human adenoviruses (CanSino) are engineered non-replicating virus vectors used to transmit genetic information. Due to the stimulation of both B- and T-cell responses, these virus vectors can elicit a significant immunological response. However, some of these vectors are influenced and partially neutralized by pre-existing vector immunity, and vector immunity might be troublesome when prime-boost regimens are utilized [47,48]. Furthermore, adverse events in Chimpanzee adenovirus vaccination have been observed [49].

In addition to several vaccines currently being developed, alternative medicines should be explored to overcome the possibility of the emergence of more powerful mutated viruses that could evade the immune system, which may lead to multiple shots of vaccination required and eventually vaccines’ failure.

## 6. Efficiency of Antiviral Treatment in COVID-19

The development of an effective antiviral treatment against SARS-CoV-2 is critical to minimize the negative impact of COVID-19 globally. Remdesivir, a nucleotide analogue, appears to show favorable results against COVID-19 by preventing viral entry and replication. Apart from Remdesivir, Favipiravir was also proven effective against SAR-CoV-2 via mechanisms similar to Remdesivir, while Favipiravir proved to be useful in mild-to-moderate cases of COVID-19, whereas Remdesivir is more effective in more severe forms of COVID-19 [50]. However, the use of Remdesivir on a large scale is limited due to its high cost and low accessibility for most patients [51,52,53,54].

Additionally, antiviral therapy with Lopinavir/Ritonavir (LPV/r) demonstrated some favorable results in improving blood oxygen saturation and reduced the days of hospitalization among COVID-19 patients as compared with Darunavir/Ritonavir (DRV/r) [55]. Nonetheless, LPV/r therapy does not improve clinical outcomes, mortality rate, time to reverse transcription polymerase chain reaction (RT-PCR) negativity, or chest CT clearance [56,57]. Currently licensed antiviral drugs used in COVID-19 patients include molnupiravir (a nucleoside analogue), Nirmatrelvir (a SARS-CoV-2 main protease inhibitor), and Ritonavir (an HIV-1 protease inhibitor), where they have demonstrated greater risk reduction in hospitalization and death [58].

WS sharing almost similar mechanisms by acting as a protease inhibitor might be another alternative against SARS-CoV-2 [59]. Moreover, the herbals play a pivotal role in combating COVID-19, owing to their antiviral effects and ability to boost the immune response contributed by antioxidant and anti-inflammatory properties, making the phytoconstituents present in herbs a promising approach to develop antiviral drugs and the beneficial effect of consuming it raw or in extract form to some extent [60]. Additionally, the use of herbal products as antiviral therapy might offer multiple mechanisms of action by targeting different crucial molecular proteins that are prominent for viral attachment, entry, and replication, as well as host cell target proteins [61].

Some other drugs that have been used in COVID-19 patients include Aviptadil, which showed some improvements in patients’ life expectancy by optimizing oxygenation and modulating cytokine storm in COVID-19–induced respiratory failure [62]. Meanwhile, the use of Hydroxychloroquine was authorized for emergency use in COVID-19 started on 28 March 2020. By 24 April 2020, the Food and Drug Administration (FDA) issued a Drug Safety Communication warning about potentially fatal prolongations of the QT interval in electrocardiograms and risks of other serious cardiac arrhythmias caused by Hydroxychloroquine [63,64,65,66,67]. The available antivirals against SAR-CoV-2 are summarized in Table 1.

## 7. *Withania somnifera* and Its Traditional and Medicinal Uses

*Withania somnifera* (WS), also known as Ashwagandha in the Indian medicinal system, is known as Ayurveda and an important “Rasayana” herb from the family *Solanaceae*. WS is widely distributed worldwide and generally found in drier regions of the world, ranging from southwest Asia to northern Africa and the Mediterranean [68]. It is also referred to as “Indian winter cherry” or “Indian Ginseng” and has been used in Ayurveda for its wide-ranging health benefits. Rasayana is a metallic or herbal preparation that promotes vitality and youthfulness besides improving physical and mental well-being. Generally, Rasayana herbs are given to young children as tonics and the elderly to increase their longevity. WS holds a prominent place among all the Rasayana herbs cited in Ayurveda. Additionally, it also holds an important place in Chinese traditional medicine and the Unani system of medicine. It is commonly available as churna (finely sieved powder), commonly consumed as a mixture with water, honey, or clarified butter (ghee). Classical literature shows that, based on its content of steroidal alkaloids, some varieties are superior to others and found that fresh Withania powder provides maximum benefits [69].

Apart from its crude extracts, the primary phytochemicals of WS (withanolides) have shown promising therapeutic properties. The most important biologically active phytochemicals of WS are steroidal lactones (e.g., withanolides and withaferins), alkaloids (e.g., anaferine, isopelletierine, and anahygrine, etc.), saponins, and some glycosides, such as sitoindosides and acylsterylglucosides [69]. WS leaves are classically used as bitters for fever and painful swellings, while flowers are used as diuretics and aphrodisiacs due to their astringent taste. Its seeds are commonly used as anthelmintics. Memory loss, syncope, obesity, hysteria, anxiety, and low sperm count are also some of the other popular applications of WS in folklore [70]. WS extracts were proposed to decrease infertility among male subjects due to the enhanced semen quality contributed by promoting enzymatic activity in seminal plasma and decreased oxidative stress [71]. Although some animal studies have found WS has reversible spermicidal and infertilizing effects in male individuals, WS extracts can also improve luteinizing hormone and follicular-stimulating hormone balance, leading to folliculogenesis and increased gonadal weight [71]. Thus, WS was discovered to enhance spermatogenesis and sperm-related indices in males and sexual behaviors in females [71].

Additionally, the traditional use of WS for bronchitis, malaria, and Alzheimer’s disease has also been reported in the earlier Egyptian literature. Research reports based primarily on preclinical studies and a few clinical trials have emphasized the neuroprotective role of WS against many neurodegenerative diseases, including Alzheimer’s, Huntington’s, and Parkinson’s diseases. The protective effects of WS were accomplished by reestablishing mitochondrial and endothelial function and mitigating apoptosis, inflammation, and oxidative stress mechanisms [72]. WS was also reported to exert strong cardioprotective effects in isoprenaline-induced myonecrosis in rats attributed by augmentation of endogenous antioxidants, maintenance of the myocardial antioxidant status, and significant restoration of most of the altered hemodynamic parameters [73].

Additionally, Ethiopians have used WS for epilepsy, anthrax, and asthma, while South African people have used it for contraception, kidney diseases, and skin infections [74]. Although most WS pharmacological activities are attributed to two main withanolides, namely withaferin A and withanolide D, other phytochemicals are also used in less proportion. WS has also demonstrated good anticancer effects by targeting cancer cell proliferation along with its anti-inflammatory effects to provide double advantage in cancer therapy [75]. Owing to its anti-inflammatory properties, WS has been suggested as a valuable supplement to ameliorate human arthritis as demonstrated in collagen-induced arthritic (CIA) rats supplemented with aqueous extract of WS roots [76].

## 8. Therapeutic Effects and Associated Molecular Mechanisms of *Withania somnifera* and Its Phytochemicals against COVID-19

In addition to the above-mentioned pharmacological properties and wide use in traditional medicine, WS has also exhibited promising antiviral potentials against SARS-CoV and SARS-CoV-2 [77]. Some other reviews also emphasized the potential antiviral activity against SARS-CoV-2 [78,79]. Furthermore, an interim analysis by Chopra, et al. [80] suggested that WS could be a safer chemoprophylaxis than the currently available disease-modifying drug, hydroxychloroquine. In this section, information about the therapeutic effects and associated molecular mechanisms of WS crude extracts and their phytochemicals against SARS-CoV-2 infection are described.

### 8.1. Antiviral Potentials of WS Crude Extracts against SARS-CoV-2

The immune system plays a critical role in protecting our body from infections by producing antibodies to attack foreign microorganisms. WS is believed to boost the immune system of people at risk of infection and during widespread SARS-CoV-2 infections, as it significantly improved the immune profile of healthy subjects by manipulating innate and adaptive immune systems [81]. Moreover, a systematic review revealed that the immunomodulatory activity of WS could be used as a prophylaxis and treatment against SARS-CoV-2 infection, as it promoted the body’s defense mechanism to improve health and longevity [82]. Additionally, an in vivo study found that WS aqueous extract promoted the immunomodulatory properties of mice along with the enhancement of antioxidant properties of milk when both of them were taken simultaneously [83]. Similarly, WS root extracts could also activate lymphocytes when administered with whole milk in human peripheral blood samples, thus boosting immune response [84]. These properties thus encourage the immune system to eradicate harmful pathogens from the body.

In SARS-CoV-2 patients, the release of immune cells’ inflammatory cytokines increases the gaps between endothelial cells along the blood vessels. Consequently, it stimulates vascular leakage that leads to the infiltration of inflammatory cells [85]. Intriguingly, WS aqueous extract is known to inhibit histamine-induced endothelial cell contraction, which may inhibit venular intercellular gaps [59]. An immunological study of WS root extract in HIV patients illustrated a significant reduction of CD38-expressing cytotoxic T lymphocytes (CD8^+^ T cells) [86]. It further showed that WS root extract could strongly activate macrophage functions to enhance the secretion of nitrite, IL-12, and TNF-α [86]. Meanwhile, WS root and leaf extracts were shown to significantly stimulate T helper type 1 (Th1) immunity and promoted the secretion of interferon-gamma (IFN-γ) and IL-2 [87]. In parallel, it enhanced the proliferation of CD4^+^/CD8^+^ and NK cells together with an increased expression of CD40/CD40L/CD80 [87]. These findings might be beneficial in combating intracellular pathogens, including SARS-CoV-2, and managing immune-suppressed diseases.

Based on the current findings, WS was found to inhibit SARS-CoV-2 infection by targeting SARS-CoV-2 structural proteins [77]. Besides, WS has shown to be associated with the ion channel activity of the active inhibitor E protein by directly binding to the pore region, which directly causes the inhibition of virus replication [88]. Additionally, WS may provide symptomatic relief of fever, one of the early symptoms of SAR-CoV-2 infection, because it also demonstrated antipyretic effects by inhibiting cyclooxygenase-2 (COX-2) enzymes [89]. WS has a long history of use for its antiviral properties for the common cold and fever [90]. Thus, it might be a promising choice in SARS-CoV-2 management, either as a prophylaxis, primary symptomatic treatment, or an alternative drug.

### 8.2. Antiviral Potentials of Phytochemicals of Withania somnifera against SARS-CoV-2

WS contains several bioactive phytochemicals, with steroidal lactones being the predominant class. The predominant examples of steroidal lactones are withanone, withanolides (particularly A and D), and withaferin A [91]. Meanwhile, several withanolide glycosides in WS, such as withanosides I, II, III, IV, V, VI, and VII, were identified by Matsuda et al. [92]. Among these, withanolides, particularly withaferin A, withanoside V and X, and withanone (Wi-N), might have potential roles in reducing the severity of SARS-CoV-2 and are useful in treating COVID-19 patients [91]. Several other phytochemicals, such as withaferin A, have shown their potential to inhibit SARS-CoV-2 infection. The chemical structure of relevant WS phytochemicals exhibiting anti-SARS-CoV-2 properties is shown in Figure 5. The summative information on their therapeutic effects against SARS-CoV-2 infection and associated molecular mechanisms is shown in Table 2 and Figure 6.

#### 8.2.1. Withaferin A

The SARS-CoV-2 infection could lead to a potentially life-threatening condition resulting from the excessive release of pro-inflammatory cytokines into the blood that gives rise to a cytokine storm [107]. Relatedly, anti-inflammatory evidence of withaferin A has been shown in ovarian cancer cells by suppressing the secretion levels of various pro-inflammatory cytokines, such as tumor necrosis factor-alpha (TNFα), IL-6, IL-8, and IL-18 in ovarian cancer cells [108]. Maurya et al. [93] predicted that WS phytochemicals, namely withaferin A, withanolide B, withanone, and somniferin showed the capacity to inhibit inflammatory mediators. This is because it showed a significant binding affinity and inhibitory activity of the inflammatory mediators, such as COX-2, phospholipases A2 (PLA2), NF-κB-inducing kinase (NIK), and interleukin-1 receptor-associated kinase 4 (IRAK-4). However, no interaction between NIK and withanone was found [93]. This anti-inflammatory effect might benefit SARS-CoV-2 patients, at least preventing the development of cytokine storms.

Besides, Sudeep et al. [96] reported the inhibitory activity of glucose-regulated protein 78 (GRP78), a chaperone-like protein found in the endoplasmic reticulum, and SARS-CoV-2 M^pro^ by withaferin A thus prevents viral entry and viral replications. Consequently, this might alleviate SARS-CoV-2 infection and the spread of the infection. Apart from inhibiting SARS-CoV-2 entry, withaferin A was also found to inhibit SARS-CoV-2 replication by inhibiting SARS-CoV-2 main protease (M^pro^) [90]. Similarly, another study also demonstrated inhibition of SARS-CoV-2 main protease M^pro,^ hence, disrupting SARS-CoV-2 replication by withaferin A along with withanone [94].

Meanwhile, a review study by Straughn and Kakar emphasized that withaferin A treatment alone or in combination with other drugs, such as hydroxychloroquine and dexamethasone, could potentially inhibit SARS-CoV-2 infection and spread, owing to its anti-tumorigenicity and capability of binding to the S-protein of SARS-CoV-2 [109]. These findings indicate that withaferin A could be a promising future lead candidate against SARS-CoV-2 in view of its therapeutic activity. However, its potential to target other structural proteins of SARS-CoV-2 or receptors or proteins mediating their replication in host cells still requires further investigation.

#### 8.2.2. Withanolide A

Srivastava et al. [97] evaluated selected phytochemicals of WS, which was extracted from leaves, stems, roots, and flowers, including withaferin A, withanolide A, withanolide B, withanolide D, withanolide E, withanone, viscosalactone B, anaferin, and withasomnine. Most WS phytochemicals displayed a significant strong binding affinity to human ACE2 receptors, the SAR-CoV and SARS-CoV-2 S protein, and the two main SARS-CoV-2 proteases, namely the primary viral proteinase (3CL-pro) and papain-like protease (PL-pro). Among the tested phytochemicals, withanolide A exhibited a strong binding affinity and inhibitory activity to SARS-CoV and SARS-CoV-2 S protein, SARS-CoV 3CL-pro main protease, and SARS-CoV-2 NSP10/NSP16 complex [97]. Cumulatively, these activities inhibited viral replication and recognition by host cells. Additionally, it was also found that withanolide A, withanolide B, and withanone were the most effective WS phytochemicals because of their binding energy [97]. Taken together, this study concluded that WS phytochemicals, particularly withanolide A, could serve as a SARS-CoV-2 inhibitor by impeding viral entry and replication.

#### 8.2.3. Withanolide R

According to Parida et al. [98], among eight phytoconstituents of WS screened, withanolide R demonstrated the most potent main protease (NSP5) inhibitor, consequently inhibiting the survival of SARS-CoV-2. This is because NSP5 is one of the pivotal enzymes essential for viral replication and transcription. Therefore, withanolide R could be a potential anti-SARS-CoV-2 drug. However, further in vivo experimental evaluation and clinical validation are required to determine its suitability as a therapeutic agent in combating SARS-CoV-2, as well as its potential targeting therapeutic effects against other structural proteins and NSPs.

#### 8.2.4. 2,3-Dihydrowithaferin A

In addition to withanolide R reported by Parida et al. [98], the same group also found that 2,3-Dihydrowithaferin A displayed the most potent binding affinity to S protein as compared with the other seven WS phytochemicals that were being studied. Hence, the inhibition of S protein by 2,3-Dihydrowithaferin A could prevent SARS-CoV-2 from entering host cells. It might be another potential phytochemical exhibiting inhibitory activity against SARS-CoV-2 that is noteworthy to evaluate further using both in vitro and in vivo studies before successfully translating to clinical applications.

#### 8.2.5. Withanone

Withanone is one of the derivatives of WS with promising pharmacological properties. Withanone and withaferin A were predicted to prevent SARS-CoV-2 entry by binding to the TMPRSS2 receptor [95]. However, withanone had a relatively stronger binding affinity to TMPRSS2 than withaferin A. Additionally, the study discovered that withanone exerted a dual mechanism to inhibit viral entry in SARS-CoV-2-infected cells by downregulating TMPRSS2 mRNA expression and decreasing binding to TMPRSS2 [95]. Besides, Patil et al. [99] revealed that withanone also could inhibit the SARS-CoV-2 main protease 3clpro and S protein. While Dhanjal et al. [100] reported that withanone possessed potent inhibitory activity against TMPRSS2 and M^pro^, together with withanoside V and methoxy withaferin A. Similarly, Kumar et al. [101] also demonstrated that withanone could inhibit M^pro^. However, their study showed no significant binding affinity or inhibitory activity by withaferin A, concluding that withanone is superior to withaferin A in targeting M^pro^ [101]. Given that M^pro^ is a SARS-CoV-2 key enzyme that has a crucial role in mediating viral replication and transcription [110], thus inhibiting M^pro^ can effectively inhibit viral replication and transcription in host cells.

Other than the above, withanone was also found to significantly attenuate the interaction between the host ACE2 receptor and RBD located on the SARS-CoV-2 S-protein [102]. In addition to the ACE2 receptor, GRP78 has also been reported to facilitate viral entry, where it translocates to the cell surface under stress. GRP78 promoted viral entry through the substrate-binding domain (SBD) [96]. Moreover, it was predicted to interact with SARS-CoV-2 S protein [111]. Thus, inhibiting the interaction between the SARS-CoV-2 S protein and GRP78 receptor on host cells could likely reduce viral infection rates. Apart from that, a study showed that withanone and other WS phytochemicals, such as withaferin A, withanolide A, withanolide B, and withanolide D, could interact with SARS-CoV-2 M^pro^ and structural proteins (i.e., S, E, and C) to inhibit viral translation, replication and viral entry into host cells [93].

#### 8.2.6. Withanoside IV

Vuai et al. [103] reported that withanoside IV and withanoside V isolated from the root of WS showed potent inhibition against NSP10 of SARS-CoV-2, a protein responsible for the methylation of NSP16. Notably, dewetting or displacing the water molecule into bulk solvents of the NSP10 receptor would provide a stable interaction of withanoside IV and NSP10 residues. Withanoside IV also could stabilize NSP10-NSP16 interaction by forming hydrogen bonds with NSP16 residues. The interaction between the NSP10-NSP16 complex subsequently abrogated NSP10-NSP16 switching mechanisms in SARS-CoV-2, thus resulting in its fatality [103]. Therefore, this finding attains the milestone of developing anti-SARS-CoV-2 from WS phytochemicals, particularly withanolide IV. Moreover, the mutation and variation of SARS-CoV-2 have been a great concern, as they will probably affect the effectiveness of drug therapy and vaccine. In this regard, targeting NSP, particularly NSP10, might be important in addressing viruses prone to mutation and variation, including SARS-CoV-2. This is because NSPs have a well-conserved genetic sequence [103].

#### 8.2.7. Withanoside V

Out of 40 WS phytochemicals, Tripathi et al. [105] discovered four phytochemicals had the highest docking energy, including withanoside II, withanoside IV, withanoside V, and withanoside IX. Among these, they found that withanoside V might be a potential drug to be developed as anti-SARS-CoV-2 because it exhibited a strong binding affinity and high stability in the protein active site of M^pro^ [105]. Besides, Shree et al. [104] also suggested the possible inhibitory activity and high affinity of withanoside V against SARS-CoV-2 M^pro^, with similar effects also found in somniferin, which is also a WS phytochemical. Apart from targeting M^pro^, withanoside V was also found to inhibit TMPRSS2, thereby inhibiting viral entry into host cells [100]. Therefore, these findings suggest the potential SARS-CoV-2 inhibitory activity of withanoside V in preventing viral recognition, entry, and replication, potentially adding it to the list of the potential drugs to be developed for managing SARS-CoV-2.

#### 8.2.8. Withanoside X

Chikhale et al. [106] evaluated the potential of withanoside X isolated from WS in inhibiting SARS-CoV-2 host entry and viral replication. The results showed a significant interaction of withanoside X with SARS-CoV-2 NSP15 endoribonuclease and S protein. NSP15 endoribonuclease is conserved across coronaviruses and is responsible for processing viral RNA, including SARS-CoV-2, to evade recognition by host immune systems [112]. However, SARS-CoV-2 evolution has been a global challenge to introducing VOC or alterations that affect virus features, such as transmissibility and antigenicity, most likely in response to the changing immunological profile of the human population [113]. Therefore, developing a drug that targets NSP15 could be considered in view of the potential mutation of SARS-CoV-2 to prevent treatment failure.

## 9. Identification of Anti-SARS-CoV-2 Inhibitors from *Withania somnifera* Using an In Silico Molecular Docking Approach

Understanding the interaction of SARS-CoV-2 S protein with cell-surface receptors is critical for developing therapeutic approaches to combat the COVID-19 pandemic. SARS-CoV-2 has been shown to interact with ACE2 receptors by binding to S protein’s RBD. The ACE2 receptor allows SARS-CoV-2 to infect human cells [114]. For instance, Balkrishna et al. [102] used molecular docking to screen thousands of WS phytochemicals against the ACE2-RBD complex, performed MD simulations, and calculated the electrostatic component of binding free energy, as well as salt bridge electrostatics. The study revealed that withanone effectively docked in the binding interface of the ACE2-RBD complex (Table 3), and it moved toward the interface center on simulation. Besides, withanone was also found to significantly reduce the electrostatic component of the ACE2-RBD complex’s binding free energy. At the interface, two salt bridges were discovered, but the addition of withanone destabilized these salt bridges and reduced their occupancies. As a result of the disruption of electrostatic connections between the RBD and ACE2, it could be hypothesized that SARS-CoV-2 entry and subsequent infectivity would be blocked or weakened.

The current in silico analysis reveals the presence of a SARS-CoV-2 S protein-GRP78 binding site, paving the way for pharmaceutical companies to create effective inhibitors to prevent the binding and hence the infection [101]. GRP78 may translocate to the plasma membrane in response to cell stress [115,116]. Once localized on the cell surface, the cell membrane became vulnerable to viral recognition via the SBD and allowed viral entry. A computational approach was used to predict the binding pattern of SARS-CoV-2 S protein with GRP78-SBD [111]. Sudeep et al. [96] docked withaferin A against GRP78 (PDB ID: 5E84) and found the binding energy to be −8.7 kcal/mol, indicating the strongest interaction as compared to the other tested compounds (Table 3).

The main protease of the coronaviruses (Mpro/3CLpro) is another important molecular target in developing anti-CoV medication due to its role in mediating viral replication and infection [117]. HCoV M^pro^ is a cysteine protease involved in the proteolytic activity of viral polyprotein cleavage [118,119]. The crystal structure of the coronavirus M^pro^ has recently been solved and made public in the Protein Data Bank. SARS-CoV-2 M^pro^ is essential for disease propagation because it processes the polyproteins needed for replication. As a result, it denotes a significant drug discovery target. Using a computational technique, Tripathi et al. [105] assessed the potential of 40 WS natural chemical constituents to find a putative inhibitor against the major protease of SARS-CoV-2. The docking study discovered that four WS phytochemicals exhibited the highest binding energy, including withanoside II (−11.30 Kcal/mol), withanoside IV (−11.02 Kcal/mol), withanoside V (−8.96 Kcal/mol), and sitoindoside IX (−8.37 Kcal/mol) (Table 3). A 100-ns simulation of molecular docking predicted withanoside V had a high binding affinity and hydrogen-bonding interactions with the active region of proteins, demonstrating its stability. Compared to other phytochemicals, the binding free energy score correlated with the highest score of −87.01 ± 5.01 Kcal/mol. The study concluded that withanoside V may be a possible inhibitor of SARS-CoV-2 M^pro^ to combat devastating COVID-19 and may have an antiviral effect on SARS-CoV-2.

A better binding affinity of WS phytochemicals to the targeted proteins was based on the binding energy and number of hydrogen bond interactions [120]. This analysis will then provide better stability of interactions. For instance, among withaferin A small derivative, withaferin A molecule 61 formed five polar H-bonds with the SARS-CoV-2 main protease (M^pro^), where amino acids involved are GLU166, THR190, CYS145, MET165, and GLN152 [90]. Consequently, out of three phytocompounds being studied, withaferin A is the best candidate to be developed as a potential inhibitor against M^pro^ [27]. Correspondingly, the study of 73 withanolides illustrated the satisfying conformations with the SARS-CoV-2 M^pro^ by forming stable complexes resulting from a more significant number of hydrogen bonds and van der Waals interactions than the complexes with reference compounds [121].

Apart from that, the hydrophobic properties of WS phytochemicals provide added value in drug products. For instance, among 25 WS phytochemicals, four best compounds, namely CID 10,100,411, CID 3,035,439, CID 101,281,364, and CID 44,423,097, illustrated the inhibition of SARS-CoV-2 replication by directly binding to the hydrophobic pore region of the E protein, thereby blocking its channel activity due to their high hydrophobic properties of best compounds [88]. Additionally, hydrophobicity properties can also improve the drug bioavailability, as demonstrated by withanolide Q predicted to possess the highest bioavailability properties owing to the highest number of proteins and drug-likeness score [120]. The properties of withanolide Q could then be a guide in developing an effective drug formulation to improve bioavailability.

## 10. Strategies for Efficient Delivery of *Withania somnifera* and Its Phytochemicals

Due to WS’s great therapeutic activity, a suitable delivery system should be explored to enhance its therapeutic efficacy and targeted delivery. WS could be a promising drug against SARS-CoV-2 taking into account the vast availability of WS in the form of liquid, tablet, and powder in India and might facilitate researchers to formulate a suitable dosage form of WS [122].

Intranasal delivery may be considered a useful and reliable route of administration in view of the potential therapeutic response because the ACE2 receptor is predominantly distributed on nasal cells, the mucosal surface along the respiratory tract, and surrounding the eyes [77]. Besides, ACE2 is one of the receptors that has been inhibited by WS phytochemicals to prevent SARS-CoV-2 viral entry into host cells [123]. Moreover, intranasal delivery could bypass the hepatic first-pass metabolism; consequently, it may provide a more rapid onset of action and rapid mucociliary clearance [120]. Even though remdesivir is an antiviral that has shown significant clinical improvement in SARS-CoV-2 patients, a detailed observation of the relationship between the drawbacks of remdesivir therapy includes inadequate pulmonary distribution and hepatotoxicity [124]. However, WS was found to minimize the hepatotoxicity of nebulizer and inhaled remdesivir in SARS-CoV-2 patients when both drugs are co-administered [124]. Thus, WS might not only act as an antiviral by itself but also minimize the side effects of other antivirals.

WS crude extracts could be encapsulated using niosomes and solid lipid nanoparticles (SLNs) delivery vesicles and release bioactive phytochemicals, including withaferin A and withanolide A to certain layers of the skin [125]. Transdermal gel (proniosomes) could also be a suitable delivery system to overcome skin barriers. Besides, Tiwari et al. [126] proposed using a super disintegrating agent to design the delivery system to facilitate the management of the dosage regimen of water-soluble granules of WS. Topical application provides a simple and convenient drug application and, therefore, could improve drug adherence in drug therapy.

A study by Monika and Munish [127] suggested that proniosomes formulation of WS leaf extract was highly stable and could be a promising delivery system, as it showed a prolonged in vitro drug release (60.8% over 24 h), as well as demonstrating an anti-inflammatory response. Meanwhile, it was discovered that phytochemicals in WS leaf extract contained withanolide A, withanolide B, and withaferin A, which could be formulated as biodegradable poly(ε-caprolactone) (WSE-PCL) and methoxy poly(ethylene glycol)-poly (e-caprolactone) (WSE-MPEG-PCL) [128]. WSE-PCL and WSE-MPEG-PCL demonstrated a better protective effect against oxidative damage induced by tert-butyl hydroperoxide (t-BHP) as compared to free WSE [128]. Marslin et al. [128] proposed WSE-MPEG-PCL nanoparticles as an efficient drug delivery method, especially for brain delivery, because they showed the highest neuroprotection effect.

The study of developing a suitable and efficient delivery system for other parts of WS, particularly root extract, could also be explored to protect phytochemicals from oxidative damage, improve their bioavailability, and develop a convenient route of administration to improve adherence to medications. Taken together, the encapsulation of WS phytochemicals necessitates and requires further investigations of its value in helping the management of preventing and treating SARS-CoV-2 infection.

## 11. Conclusions and Future Perspectives

The present review was explicitly focused on WS, which is considered one of the important herbal medicine candidates against SARS-CoV-2. Different aspects of understanding the pharmacological role, molecular mechanism, in silico molecular docking, and efficient delivery of WS and its phytochemicals were outlined. It was found that the key receptor in SARS-CoV-2 pathogenesis was ACE2, which attaches to the S protein before entering into host cells. Prior to this interaction, the S protein is activated first by GRP78 and TMPRSS2. Notably, WS phytochemicals are found to competitively block these three receptors, preventing the SARS-CoV-2 entry into host cells. In SARS-CoV-2-infected patients, the viruses already invaded host cells and consequently replicated in a more significant amount using host energy and molecular substrates. The replication of SARS-CoV-2 is contributed by the main proteases responsible for the maturation of SARS-CoV-2. Intriguingly, WS phytochemicals are also found to target main proteases, including 3CL-pro, M^pro^, and NSP5. Furthermore, withanolide A and withanoside IV demonstrated inhibitory activity of SARS-CoV-2 NSP10-NSP16, resulting in the inhibition of SARS-CoV-2 recognition and replication, thus inducing fatality of SARS-CoV-2. WS phytochemicals also can inhibit SARS-CoV-2 NSP15 endoribonuclease, which is contributed by withanoside X and quercetin glucoside. This finding suggests multiple SARS-CoV-2 inhibitory mechanisms by WS phytochemicals, owing to the prevention and treatment of SARS-CoV-2 activity. Moreover, WS also possesses immunomodulatory, anti-inflammatory, and antioxidant properties, which might be beneficial in SAR-CoV-2 patients to prevent exacerbation of COVID-19 severity.

Since COVID-19 mortality is unpredictable and has risen sharply, modern medicine and therapy dependence have risen. At present, nanotechnology approaches should be emphasized as nanotechnology might provide important benefits and prevent drawbacks of drug products. This is because nanotechnology in drug products offers advantages, such as enhancing drug delivery through impermeable barriers, thereby improving the efficacy and antiviral delivery of WS and its phytochemicals. The efficient delivery could increase the active targeting of WS and its phytochemicals against SARS-CoV-2 proteins and host cell receptors/proteins, consequently decreasing the emergence of drug resistance. Apart from that, nanoformulations might provide drug protection against deactivation, enhance stability and bioavailability, enable controlled drug release to the target through engineered moieties, increase specificity toward drug targets, and lower the incidence of adverse effects.

Importantly, the discovery of the targeting site of WS and its phytochemicals against SARS-CoV-2 structural and NSPs, as well as auxiliary proteins and proteins/receptors in host cells, is required to further understand the antiviral potentials of WS, which could provide deep scientific evidence or baseline about this plant and its phytochemicals for developing highly efficacious anti-SARS-CoV-2 drugs. Other than these, using in silico to predict the drug-likeness and structure-binding and structure-activity relationships between WS phytochemicals with targeting proteins in SARS-CoV-2 should be applied widely to secure research funding for selecting the promising phytochemicals against SARS-CoV-2 using preclinical models, as well as supporting data or validating preclinical data. Nowadays, state-of-art artificial intelligence has empowered the knowledge of computer engineers and health scientists to closely ally in improving respiratory disease prognosis and accurate drug formulation with the available herbal compounds. Moreover, the ability of new drugs to target the current issues with clinical and biomedical analyses is highly favorable for designing an artificial intelligence-based model in clinical drugs. With that concern, the augmentation of new pharmaceutical formulations and artificial intelligence with in silico models for drug compounds are highly recommended for future viral therapy.

Lastly, one potential aspect to be explored in the near future to further understand WS’s therapeutic mechanisms and its phytochemicals in both SARS-CoV-2 and host cells is the regulation of non-coding RNAs, particularly long non-coding RNAs (lncRNAs). LncRNAs have been reported to play a crucial role in carcinogenesis and chemoresistance [129,130]. Recent literature also indicates the crosstalk between SARS-CoV-2 and lncRNAs [129,130]. Thus, it is necessary to evaluate the regulatory roles of lncRNAs in SARS-CoV-2 pathogenesis to identify more potent therapeutic targets to fully eradicate its infection.

## Figures and Tables

**Figure 1 microorganisms-11-01000-f001:**
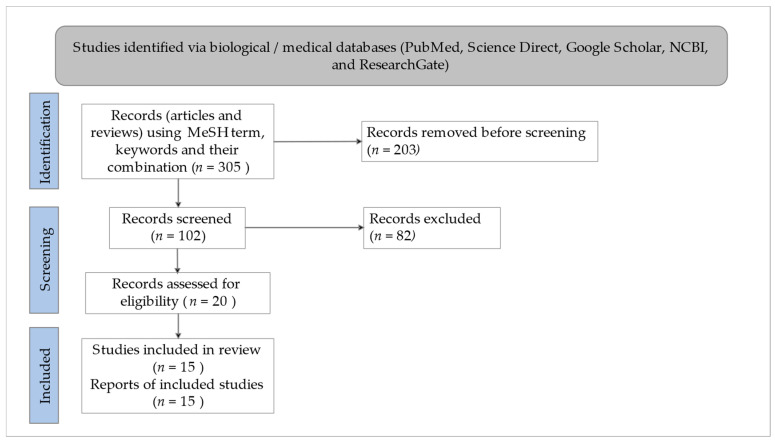
PRISMA flow chart of selecting the relevant studies.

**Figure 2 microorganisms-11-01000-f002:**
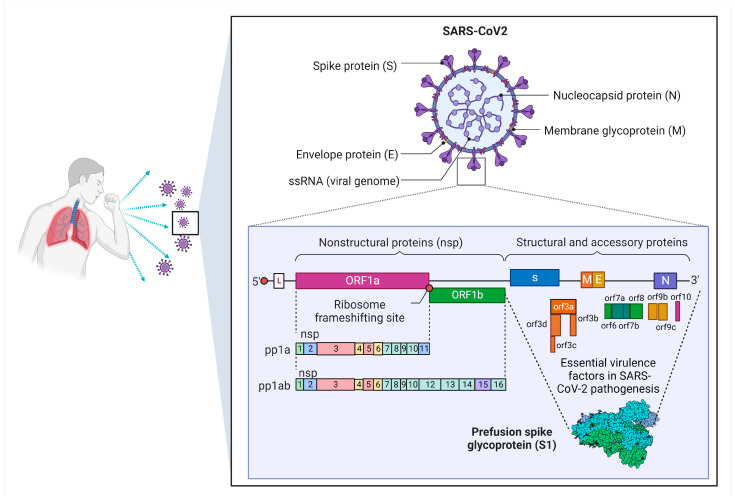
Schematic diagram of the SARS-CoV-2 genome structure. SARS-CoV-2 is spherical in shape. The virus is wrapped in a lipid envelope that is coated with spike glycoprotein. The SARS-CoV-2 genomic organization is characteristic of betacoronaviruses. The full-length RNA genome is roughly 29,903 nucleotides long and contains a 5′UTR replicase complex (composed of ORF1a and ORF1b). ORF1a is responsible for encoding NSP1–NSP10, while ORF1b is responsible for encoding NSP1–NSP16. The structural proteins are encoded by four genes: the spike gene, envelope gene, membrane gene, and nucleocapsid gene, as well as a poly (A) tail at the 3′UTR. Between the structural genes are the accessory genes.

**Figure 3 microorganisms-11-01000-f003:**
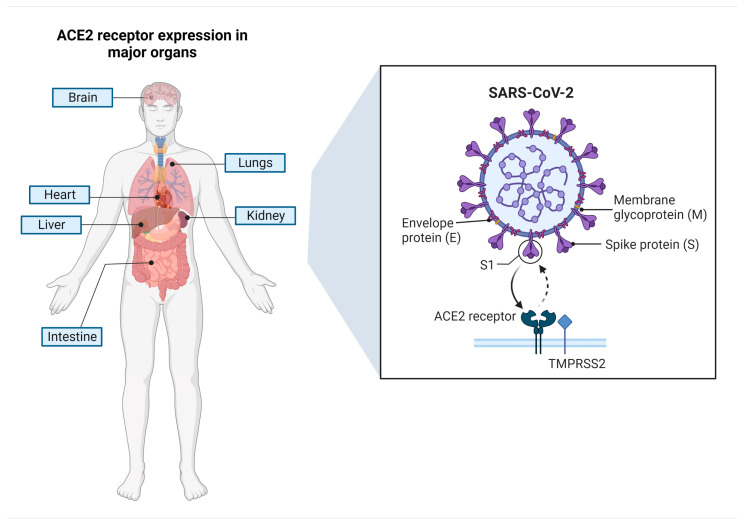
SARS-CoV-2 and host cell interaction. Major proteins, such as the spike (S), membrane (M), envelope (E), and nucleocapsid (N) proteins are indicated. The receptor (ACE2) from the host cell is displayed.

**Figure 4 microorganisms-11-01000-f004:**
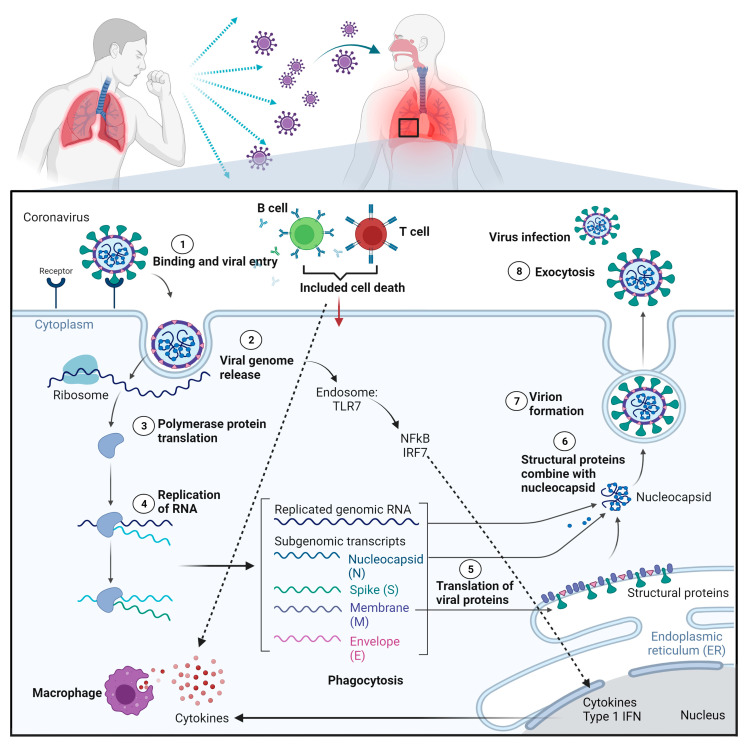
Schematic diagram representing SARS-CoV-2 pathogenesis. The surface S protein facilitates SARS-CoV-2 to enter host cells by attaching to ACE2 in synergy with the host’s TMPRSS2. In the host cells, cascade event of SARS-CoV-2 replication, leading to viral assembly, maturation, and virus release. On the other hand, SARS-CoV-2 RNA will be identified by TLR7 where multiple proteins are recruited to form a complex after viral RNA engages TLR7, which increases the transfer of transcription factors like NF-kB and IRF7 to the nucleus and triggers the release of proinflammatory cytokines that are responsible for the development of sign and symptoms of COVID-19.

**Figure 5 microorganisms-11-01000-f005:**
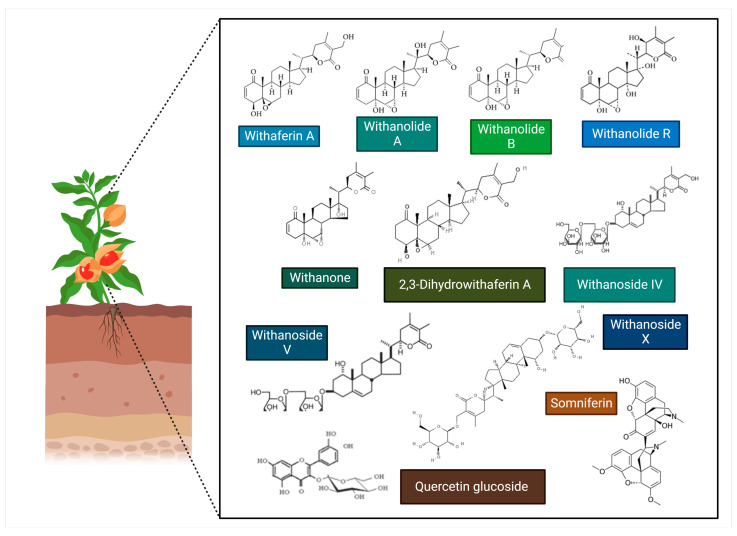
The structure of *Withania Somnifera* phytochemicals. The key phytochemicals of *Withania somnifera* include Withaferin A, Withanolide A, Withanolide B, Withanolide R, 2,3-Dihydrowithaferin A, Withanone, Withanoside IV, Withanoside V, Withanoside X, Quercetin glucoside, and Somniferin, which might exhibit SAR-CoV-2 inhibitory activity.

**Figure 6 microorganisms-11-01000-f006:**
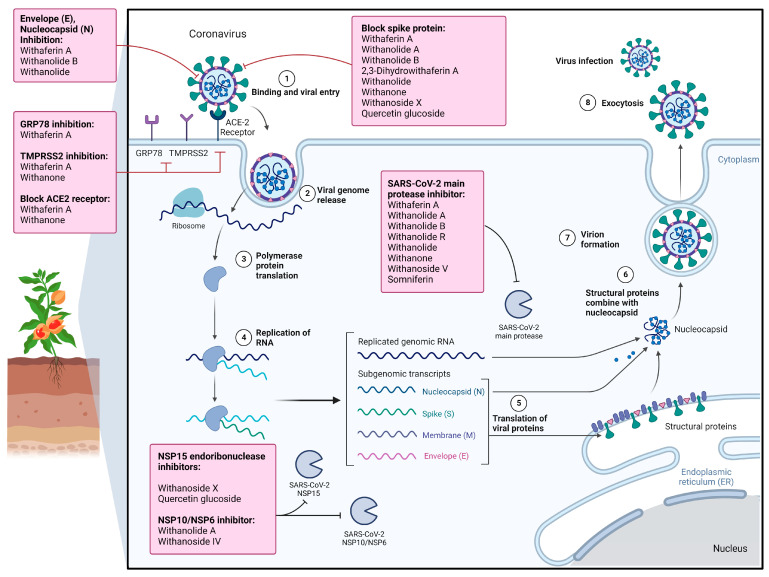
Overview of molecular mechanisms of WS phytochemicals against SARS-CoV-2. WS phytochemicals inhibit viral entry by inhibiting SARS-CoV-2 S, E, and N proteins, as well as host ACE2, TMPRSS2, and GRP78. Meanwhile, WS phytochemicals also prevent viral replication by inhibiting the viral main protease, NSP15 endonuclease, and NSP10/NSP6 protein complex.

**Table 1 microorganisms-11-01000-t001:** Antiviral treatments against SAR-CoV-2.

Antiviral	Mechanism of Action	Adverse Effects/Issue	Reference
Remdesivir	Inhibition of virus entry (pre-cytokine storm) Inhibition of virus replication (pre-cytokine storm)	Acute respiratory failure Decreased glomerular filtration rate (increased blood creatinine level) Anemia, pyrexia Increased blood glucose level Hepatotoxicity Gastrointestinal symptoms Cardiovascular toxicity	[51,52,53,54]
Remdesivir	Inhibition of virus entry (pre-cytokine storm) Inhibition of virus replication (pre-cytokine storm)	Acute respiratory failure Decreased glomerular filtration rate (increased blood creatinine level) Anemia, pyrexia Increased blood glucose level Hepatotoxicity Gastrointestinal symptoms Cardiovascular toxicity	[50,51,52,53,54]
Lopinavir/Ritonavir (LPV/r)	Inhibition of virus entry (pre-cytokine storm) Inhibition of virus replication (pre-cytokine storm)	Does not improve clinical outcome, mortality, time to RT-PCR negativity, or chest CT clearance	[56,57]
Molnupiravir	Inhibition of viral replication	Not recommended in pregnancy, women of childbearing potential, or breastfeeding mothers	[58]
Nirmatrelvir/Ritonavir	Inhibition of viral replication	Contraindicated in people with severe hepatic or renal impairment Not recommended in pregnancy, women of childbearing potential, or breastfeeding mothers	[58]
Hydroxychloroquine	Inhibition of virus entry (pre-cytokine storm) Inhibition of virus replication (pre-cytokine storm)	Increased risk of arrhythmias Prolongation of QT interval at ECG	[63,64,65,66]

**Table 2 microorganisms-11-01000-t002:** *Withania somnifera* and its phytochemicals and associated molecular mechanisms of their therapeutic effects against SARS-CoV-2.

Phytochemical	Source	Molecular Mechanism	Reference
Withaferin A	Not Available	Inhibited SARS-CoV-2 M^pro^, thus preventing SARS-CoV-2 replication	[90]
	Not Available	Inhibited SARS-CoV-2 M^pro^ and structural proteins, including S, E, and N, thus impeding viral replication and entry into host cells	[93]
	Purchased from Sigma-Aldrich (now Merck, St. Louis, MO, USA)	Inhibited SARS-CoV-2 M^pro^, thus disrupting SARS-CoV-2 replication	[94]
	Not Available	Inhibited TMPRSS2 receptor, thereby preventing viral entry into host cells	[95]
	Not Available	Inhibited SARS-CoV-2 M^pro^ and GRP78, thus impeding viral replication and SARS-CoV-2 recognition by host cells	[96]
Withanolide A	Leaves, stems, roots, and flowers extract	Inhibited SARS-CoV and SARS-CoV-2 S protein, SARS-CoV 3CL-pro main protease, and SARS-CoV-2 NSP10/NSP-16 complex, thus decreasing viral recognition and replication in host cells	[97]
Withanolide B	Not Available	Inhibited SARS-CoV-2 M^pro^ and structural proteins, including S, E, and N.	[93]
Withanolide R	Not Available	Inhibited main protease (NSP5) of SARS-CoV-2, consequently reducing the survival of SARS-CoV-2	[98]
2,3-Dihydrowithaferin A	Not Available	Inhibited SARS-CoV-2 S protein and acted as the most potent S protein inhibitor as compared with the other WS phytochemicals, preventing viral entry into host cells	[98]
Withanone	Not Available	Inhibited TMPRSS2 receptor and downregulated TMPRSS2 mRNA, thereby impeding SARS-CoV-2 entry into host cells	[95]
	Not Available	Inhibited SARS-CoV-2 main protease 3CL-pro and S protein, thus decreasing viral entry and replication in host cells	[99]
	Leaves and stem extract	Inhibited SARS-CoV-2 TMPRSS2 and M^pro^, thus prohibiting SARS-CoV-2 entry and replication in host cells	[100]
	Not Available	Inhibited SARS-CoV-2 M^pro^, thus halting SARS-CoV-2 replication in host cells	[101]
	Root extract	Inhibited viral entry by attenuating the interaction between the host ACE2 receptor and RBD located in the SARS-CoV-2 S-protein	[102]
	Purchased from Sigma-Aldrich (now Merck, St. Louis, MO, USA) (>98% purity)	Inhibited SARS-CoV-2 M^pro^, thus disrupting SARS-CoV-2 replication	[94]
Withanoside IV	Root extract	Abrogated NSP10-NSP16 switching mechanisms in SARS-CoV-2, thus promoting fatality of SARS-CoV-2	[103]
Withanoside V	Not Available	Inhibited SARS-CoV-2 M^pro^, thus attenuating SARS-CoV-2 replication in host cell	[104]
	Not Available	Inhibited SARS-CoV-2 M^pro^, thus attenuating SARS-CoV-2 replication in host cells	[105]
Withanoside X	Not Available	Inhibited SARS-CoV-2 NSP15 endoribonuclease and SARS-CoV-2 S protein, thus preventing SARS-CoV-2 entry and replications in host cells	[106]
Quercetin glucoside	Not Available	Inhibited SARS-CoV-2 NSP15 endoribonuclease and SARS-CoV-2 S proteins, thus reducing SARS-CoV-2 entry and replication in host cells	[106]
Somniferin	Not Available	Inhibited SARS-CoV-2 M^pro^, thus attenuating SARS-CoV-2 replication in host cells	[104]

**Table 3 microorganisms-11-01000-t003:** Binding energy of WS phytochemicals against protein targets of SARS-CoV-2 and its host cells.

Phytochemical	Target Protein	PDB ID	Binding Energy (Kcal/mol)
Sitoindoside IX	M^pro^	6LU7	−8.37
Withaferin A	ACE2-RBD	6M17	−9.1
Withaferin A	GRP78	5E84	−8.7
Withaferin A	M^pro^	6LU7	−9.83
Withanolide A	ACE2-RBD	6M17	−9.6
Withanolide B	ACE2-RBD	6M17	−9.4
Withanone	ACE2-RBD	6M17	−9.4
Withanoside II	M^pro^	6LU7	−11.3
Withanoside IV	M^pro^	6LU7	−11.02
Withanoside V	M^pro^	6LU7	−8.96

## Data Availability

Not applicable.

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
