# Peer review of "Phytochemicals of Withania somnifera as a Future Promising Drug against SARS-CoV-2: Pharmacological Role, Molecular Mechanism, Molecular Docking Evaluation, and Efficient Delivery"

_microorganisms, 2023, doi:10.3390/microorganisms11041000_

Round 1
Reviewer 1 Report
A Prisma flow-diagram need to use for description how to select the studies, articles included in the review, according to:
Page, MJ; McKenzie, JE; Bossuyt, PM; Boutron, I.; Hoffmann, TC; Mulrow, CD; Shamseer, L.; Tetzlaff, JM; Akl, EA; Brennan, SE; Chou, R.; Glanville, J.; Grimshaw, JM; Hróbjartsson, A.; Lalu, MM; Li, T.; Loder, EW; Mayo-Wilson, E.; McDonald, S.; McGuinness, LA; Stewart, LA; Thomas, J.; Tricco, AC; Welch, VA; Whiting, P.; Moher, D. The PRISMA 2020 statement: An updated guideline for reporting systematic reviews. Journal of Clinical Epidemiology 2021, 134, 178-189.
Page, MJ; McKenzie, JE; Bossuyt, PM; Boutron, I.; Hoffmann, TC; Mulrow, CD; Shamseer, L.; Tetzlaff, JM; Moher, D. Updating guidance for reporting systematic reviews: development of the PRISMA 2020 statement. Journal of Clinical Epidemiology 2021, 134, 103-112.
Reviewer 2 Report
The review is well-structured and interesting. It is complete and informative. The major subject is the information about Withania somnifera as a promising phytochemical against COVID problems. What I miss in the review is the lack of information about dynamic molecular studies and in vivo investigations. I would recommend including of some more information on these points, if available.
Reviewer 3 Report
This manuscript reviews the Phytochemicals of Withania somnifera as a Future Promising Drug against SARS-CoV-2: Pharmacological Role, Molecular Mechanism, Molecular Docking Evaluation, and Efficient Delivery. The topic is extremely large for this manuscript. The authors should make extensive improvements in order to develop properly their manuscript. Also, not being a new topic, the way of presentation and a suitable amount of data will increase the value of this manuscript. For so many authors it will be a short homework to make all the improvements. Therefore, I recommend specific and major improvements, before any consideration regarding the possible publication, at the level of literature search, editing, and structure, which are listed below:
Shape suggestions
Please check the Ethics in publication as your paper has 14!!! Authors for a paper with less than 15.5 real pages, in the MDPI format, which occupies 2/3 of a page.
Letters on the Figures should be of similar size with those on the main text. Now, they are very small, almost unreadable.
Figure 4 is blurred, please add a best quality one.
Content suggestions
Introduction. As the title is about Withania somnifera, a consistent paragraph must be added about its general presentation (not a short sentence (91-92). The ancient use of plants in traditional medicine must be emphasized (thousands of years ago) - I suggest as reference Bungau, S.G.; Popa, V.-C. Between religion and science: some aspects: concerning illness and healing in antiquity, Transylv. Rev., 2015, 26(3), 3-19. (https://www.researchgate.net/publication/286442576_Between_Religion_and_Science_Some_Aspects_Concerning_Illness_and_Healing_in_Antiquity), as to support the statement. Also, you must integrating the plant, detailing its multiple actions https://doi.org/10.3390/biomedicines8120571 , having role in dermatological diseases https://doi.org/10.3390/molecules26092407 , anticancer action https://doi.org/10.1007/s11356-020-09028-0 , etc. Check the above references and proceed consequently
L94-100. As being a Review, the aim of your research should emphasize the novelty it brings to the field, reason for choosing this topic, relevance of the topic. What is actually stated in this paragraph “provide detailed insights for a better understanding…”, “discusses the possible associated…”, “highlights the in silico molecular docking evaluation…” was already, provided, discussed highlighted in the published original papers from where you have inspired. Please reshape.
It must be added a 2nd section of Methodology for literature search/selection or a similar title. It would therefore be advisable to present the methodology for selecting bibliographic resources (databases used and the reason for choosing those data basis, types of documents, filtering results, inclusion/exclusion criteria for manuscripts: language, key words, duplicates, etc.). Moreover, have you searched (graphically) the impact of the topic on the general literature? You would have found out that there are thousands of articles/reviews in the field, much better developed.
In this new 2nd section, also add a Figure related to the Literature impact on the Withania somnifera/ SARS-CoV-2relationship and related research areas and paper types identified in medical databases/WoS using the most suitable Boolean search operators.
Detail better the virtual screening of substances used in the treatment of SARS-CoV-2 infection https://doi.org/10.1016/j.biopha.2022.113432
Add a new subsection/section where present the efficiency of antiviral treatment in COVID‑19 https://doi.org/10.3892/etm.2021.10080 , the actual drug treatment https://doi.org/10.1016/j.scitotenv.2021.152072https://doi.org/10.1007/s11356-021-17824-5 https://doi.org/10.1016/j.biopha.2022.112700 etc., versus plant based compounds https://doi.org/10.3390/biomedicines9091266 available for SARS-CoV-2. A table would be relevat, having the last column references (Refs.). I am sure that the authors will find the best way in providing it.
Chapter 5 is too poor.
Round 2
Reviewer 3 Report
The authors responded to my requests.